# CHARACTERIZING SPARSE CONNECTIVITY PATTERNS IN NEURAL NETWORKS

## ABSTRACT

We propose a novel way of reducing the number of parameters in the storage-hungry fully connected layers of a neural network by using pre-defined sparsity, where the majority of connections are absent prior to starting training. Our results indicate that convolutional neural networks can operate without any loss of accuracy at less than $0.5\%$ classification layer connection density, or less than $5\%$ overall network connection density. We also investigate the effects of pre-defining the sparsity of networks with only fully connected layers. Based on our sparsifying technique, we introduce the 'scatter' metric to characterize the quality of a particular connection pattern. As proof of concept, we show results on CIFAR, MNIST and a new dataset on classifying Morse code symbols, which highlights some interesting trends and limits of sparse connection patterns.

## 1 INTRODUCTION

Neural networks (NNs) in machine learning systems are critical drivers of new technologies such as image processing and speech recognition. Modern NNs are gigantic in size with millions of parameters, such as the ones described in Alexnet (Krizhevsky et al., 2012), Overfeat (Sermanet et al., 2013) and ResNet (He et al., 2016). They therefore require an enormous amount of memory and silicon processing during usage. Optimizing a network to improve performance typically involves making it deeper and adding more parameters (Simonyan & Zisserman, 2015; Szegedy et al., 2015; Huang et al., 2016), which further exacerbates the problem of large storage complexity. While the convolutional (conv) layers in these networks do feature extraction, there are usually fully connected layers at the end performing classification. We shall henceforth refer to these layers as *connected layers (CLs)*, of which fully connected layers *(FCLs)* are a special case. Owing to their high density of connections, the majority of network parameters are concentrated in FCLs. For example, the FCLs in Alexnet account for 95.7% of the network parameters (Zhang et al., 2016).

We shall refer to the spaces between CLs as CL *junctions* (or simply junctions), which are occupied by connections, or weights. Given the trend in modern NNs, we raise the question – "How necessary is it to have FCLs?" or, in other words, "What if most of the junction connections never existed? Would the resulting *sparsely connected layers (SCLs)*, when trained and tested, still give competitive performance?" As an example, consider a network with 2 CLs of 100 neurons each and the junction between them has 1000 weights instead of the expected 10,000. Then this is a sparse network with connection density of 10%. Given such a sparse architecture, a natural question to ask is "How can the existing 1000 weights be best distributed so that network performance is maximized?"

In this regard, the present work makes the following contributions. In Section 2, we formalize the concept of sparsity, or its opposite measure *density*, and explore its effects on different network types. We show that CL parameters are largely redundant and a network pre-defined to be sparse before starting training does not result in any performance degradation. For certain network architectures, this leads to CL parameter reduction by a factor of more than 450, or an overall parameter reduction by a factor of more than 20. In Section 2.4, we discuss techniques to distribute connections across junctions when given an overall network density. Finally, in Section 3, we formalize pre-defined sparse connectivity patterns using adjacency matrices and introduce the *scatter* metric. Our results show that scatter is a quick and useful indicator of how good a sparse network is.

## 2 PRE-DEFINED SPARSITY

As an example of the footprint of modern NNs, AlexNet has a weight size of 234 MB and requires 635 million arithmetic operations only for feedforward processing (Zhang et al., 2016). It has been shown that NNs, particularly their FCLs, have an excess of parameters and tend to *overfit* to the training data (Denil et al., 2013), resulting in inferior performance on test data. The following paragraph describes several previous works that have attempted to reduce parameters in NNs.

Dropout (deletion) of random neurons (Srivastava et al., 2014) trains multiple differently configured networks, which are finally combined to regain the original full size network. Chen et al. (2015) randomly forced the same value on collections of weights, but acknowledged that "a significant number of nodes [get] disconnected from neighboring layers." Other sparsifying techniques such as pruning and quantization (Han et al., 2016; 2015; Zhou et al., 2016; Gong et al., 2014) first train the complete network, and then perform further computations to delete parameters. Sindhwani et al. (2015) used low rank matrices to impose structure on network parameters. Srinivas et al. (2016) proposed a regularizer to reduce parameters in the network, but acknowledged that this increased training complexity. In general, all these architectures deal with FCLs at some point of time during their usage cycle and therefore, do not permanently solve the parameter explosion problem of NNs.

### 2.1 OUR METHODOLOGY

Our attempt to simplify NNs is to pre-define the level of sparsity, or connection density, in a network prior to the start of training. This means that our network always has fewer connections than its FCL counterpart; the weights which are absent never make an appearance during training or inference. In our notation, a NN will have $J$ junctions, i.e. $J + 1$ layers, with $\{N_1, N_2, \cdots, N_{J+1}\}$ being the number of neurons in each layer. $N_i$ and $N_{i+1}$ are respectively the number of neurons in the earlier (left) and later (right) layers of junction $i$. Every left neuron has a fixed number of edges going from it to the right, and every right neuron has a fixed number of edges coming into it from the left. These numbers are defined as fan-out ($fo_i$) and fan-in ($fi_i$), respectively. For conventional FCLs, $fo_i = N_{i+1}$ and $fi_i = N_i$. We propose SCLs where $fo_i < N_{i+1}$ and $fi_i < N_i$, such that $N_i \times fo_i = N_{i+1} \times fi_i = W_i$, the number of weights in junction $i$. Having a fixed $fo_i$ and $fi_i$ ensures that all neurons in a junction contribute equally and none of them get disconnected, since that would lead to a loss of information. The connection density in junction $i$ is given as $W_i/(N_i N_{i+1})$ and the overall CL connection density is defined as $\left(\sum_{i=1}^{J} W_i\right) / \left(\sum_{i=1}^{J} N_i N_{i+1}\right)$.

Note that earlier works such as Dey et al. (2017b;a) have proposed hardware architectures that leverage pre-defined sparsity to speed up training. However, a complete analysis of methods to pre-define connections, its possible gains on different kinds of modern deep NNs and a test of its limits via a metric quantifying its goodness has been lacking. Bourely et al. (2017) introduced a metric based on eigenvalues, but ran limited tests on MNIST. The following subsections analyze our method of pre-defined sparsity in more detail. We experimented with networks operating on CIFAR, MNIST and Morse code symbol classification – a new dataset described in Dey (2017)[1].

### 2.2 NETWORK EXPERIMENTS

#### 2.2.1 CIFAR

We used the original CIFAR10 and CIFAR100 datasets without data augmentation. Our network has 6 conv layers with number of filters equal to $[64, 64, 128, 128, 256, 256]$. Each has window size 3x3. The outputs are batch-normalized before applying ReLU non-linearity. A max-pooling layer of pool size 2x2 succeeds every pair of conv layers. This structure finally results in a layer of 4096 neurons, which is followed by the CLs. We used the Adam optimizer, ReLU-activated hidden layers and softmax output layer – choices which we maintained for all networks unless otherwise specified.

Our results in Section 2.4 indicate that later CL junctions (i.e. closer to the outputs) should be denser than earlier ones (i.e. closer to the inputs). Moreover, since most CL networks have a tapering structure where $N_i$ monotonically decreases as $i$ increases, more parameter savings can be

---

[1]L2 regularization is used wherever applicable for the MNIST networks. For Morse, the difference with and without regularization is negligible, while for CIFAR, the accuracy results differ by about 1%

achieved by making earlier layers less dense. Accordingly we did a grid search and picked CL junction densities as given in Table 1. The phrase 'conv+2CLs' denotes 2 CL junctions corresponding to a CL neuron configuration of $(4096, 512, 16)$ for CIFAR10, $(4096, 512, 128)$ for CIFAR100[2], and $(3136, 784, 10)$ for MNIST (see Section 2.2.2). For 'conv+3CLs', an additional 256-neuron layer precedes the output. 'MNIST CL' and 'Morse CL' refer to the CL only networks described subsequently, for which we have only shown some of the more important configurations in Table 1.

As an example, consider the first network in 'CIFAR10 conv+2CLs' which has $fo_1 = fo_2 = 1$. This means that the individual junction densities are $(4096 \times 1)/(4096 \times 512) = 0.2\%$ and $(512 \times 1)/(512 \times 16) = 6.3\%$, to give an overall CL density of $(4096 \times 1 + 512 \times 1)/(4096 \times 512 + 512 \times 16) = 0.22\%$. In other words, while FCLs would have been 100% dense with $2,097,152 + 8192 = 2,105,344$ weights, the SCLs use $4096 + 512 = 4608$ weights, which is 457 times less. Note that weights in the sparse junction are distributed as fixed, but *randomly* generated patterns, with the constraints of fixed fan-in and fan-out.

Table 1: Densities for some of our sparse networks

| Net | Junction fan-outs | CL Junction Densities (%) | Overall CL Density (%) | Net | Junction fan-outs | CL Junction Densities (%) | Overall CL Density (%) |
|---|---|---|---|---|---|---|---|
| CIFAR10 conv+2CLs | 1,1 | 0.2,6.3 | 0.22 | CIFAR10 conv+3CLs | 1,1,1 | 0.2,0.4,6.3 | 0.22 |
| | 1,8 | 0.2,50 | 0.39 | | 1,1,8 | 0.2,0.4,50 | 0.3 |
| | | | | | 1,2,16 | 0.2,0.8,100 | 0.41 |
| CIFAR100 conv+2CLs | 1,1 | 0.2,0.8 | 0.21 | CIFAR100 conv+3CLs | 1,1,1 | 0.2,0.4,0.8 | 0.22 |
| | 1,8 | 0.2,6.3 | 0.38 | | 1,1,16 | 0.2,0.4,13 | 0.39 |
| | 1,32 | 0.2,25 | 0.95 | | 1,2,32 | 0.2,0.8,25 | 0.59 |
| MNIST conv+2CLs | 1,5 | 0.1,50 | 0.29 | MNIST CL ($x = 224$) | 4,10 | 1.79,100 | 3.02 |
| | 4,10 | 0.5,100 | 0.83 | | 112,10 | 50,100 | 50.63 |
| | 16,10 | 2,100 | 2.35 | Morse CL | 512,32 | 50,50 | 50 |

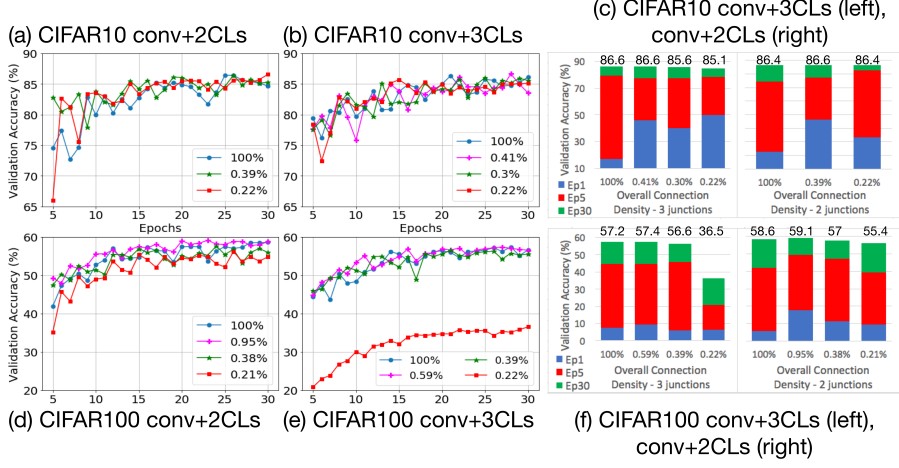

Figure 1: Performance results of pre-defined sparsity for (a)–(c) CIFAR10, and (d)–(f) CIFAR100, trained for 30 epochs using different network densities and varying number of CLs. (a),(b),(d),(e) Validation accuracy across epochs. (c),(f) Best validation accuracies after 1, 5 and 30 epochs.

Figure 1 shows the results for CIFAR. Subfigures (a), (b), (d) and (e) show classification performance on validation data as the network is trained for 30 epochs (note that the final accuracies stayed almost constant after 20 epochs). The different lines correspond to different overall CL densities. Subfigures (c) and (f) show the best validation accuracies after 1, 5 and 30 epochs for the different CL densities. We see that the final accuracies (the numbers at the top of each column) show negligible performance degradation for these extremely low levels of density, not to mention some cases where SCLs outperform FCLs. These results point to the promise of sparsity. Also

---

[2]Powers of 2 are used for ease of testing sparsity. The extra output neurons have a 'false' ground truth labeling and thus do not impact the final classification accuracy.

notice from subfigures (c) and (f) that SCLs generally start training quicker than FCLs, as evidenced by their higher accuracies after 1 epoch of training. See Appendix Section 5.3 for more discussion.

### 2.2.2   MNIST

We used 2 different kinds of networks when experimenting on MNIST (no data augmentation). The first was 'conv+2CLs' – 2 conv layers having 32 and 64 filters of size 5x5 each, alternating with 2x2 max pooling layers. This results in a layer of 3136 neurons, which is followed by 2 CLs having 784 and 10 neurons, i.e. 2 junctions overall. Fig. 2(a) and (b) show the results. Due to the simplicity of the overall network, performance starts degrading at higher densities compared to CIFAR. However, a network with CL density 2.35% still matches FCLs in performance. Note that the total number of weights (conv+SCLs) is 0.11M for this network, which is only 4.37% of the original (conv+FCLs).

The second was a family of networks with only CLs, either having a single junction with a neuron configuration of $(1024, 16)$, or 2 junctions configured as $(784, x, 10)$, where $x$ varies. The results are shown in Fig. 2(c), which offers two insights. Firstly, performance drops off at higher densities for CL only MNIST networks as compared to the one with conv layers. However, half the parameters can still be dropped without appreciable performance degradation. This aspect is further discussed in Section 2.3. Secondly, large SCLs perform better than small FCLs with similar number of parameters. Considering the black-circled points as an example, performance drops when switching from 224 hidden neurons at 12.5% density to 112 at 25% to 56 at 50% to 28 at 100%, even though all these networks have similar number of parameters. So increasing the number of hidden neurons is desirable, albeit with diminishing returns.

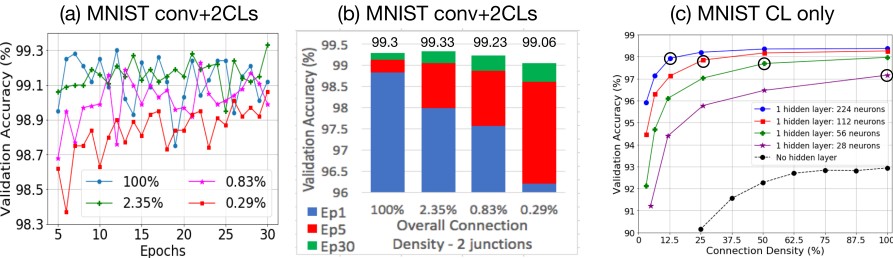

Figure 2: (a)–(b) Performance results of pre-defined sparsity on an MNIST conv network with different densities, each trained for 30 epochs. (c) Performance vs. connection density for different MNIST CL only networks, each trained for 100 epochs.

### 2.2.3   MORSE

The Morse code dataset presents a harder challenge for sparsity. It only has 64-valued inputs (as compared to 784 for MNIST and 3072 for CIFAR), so each input neuron encodes a significant amount of information. The outputs are Morse codewords and there are 64 classes. Distinctions between inputs belonging to different classes is small. For example, the input pattern for the Morse codeword '. . . . .' can be easily confused with the codeword '. . . . -'. As a result, performance degrades quickly as connections are removed. Our network had 64 input and output neurons and 1024 hidden layer neurons, i.e. 3 CLs and 2 junctions, trained using stochastic gradient descent. The results are shown in Fig. 3(a). As with MNIST CL only, 50% density can be achieved with negligible degradation in accuracy.

### 2.3   ANALYZING THE RESULTS OF PRE-DEFINED SPARSITY

Our results indicate that for deep networks having several conv layers, there is severe redundancy in the CLs. As a result, they can be made extremely sparse without hampering network performance, which leads to significant *memory savings*. If the network only has CLs, the amount of density reduction achievable without performance degradation is smaller. This can be explained using the argument of relative importance. For a network which extensively extracts features and processes its raw input data via conv filters, the input to the CLs can already substantially discriminate between

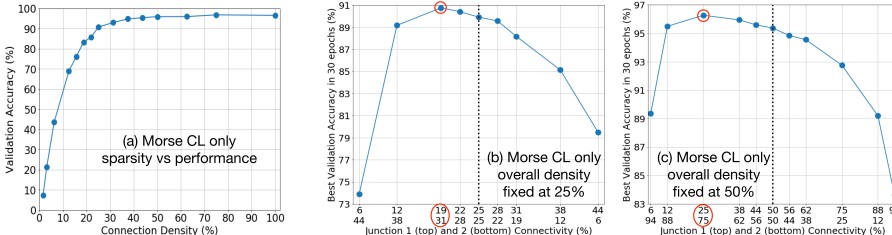

Figure 3: (a) Performance vs. connection density for a Morse CL only 2 junction network. (b) and (c) Performance results by varying individual junction densities while overall density is fixed at (b) 25% (c) 50%. All cases trained for 30 epochs.

inputs belonging to different classes. As a result, the importance of the CLs' functioning is less as compared to a network where they process the raw inputs.

The *computational savings* by sparsifying CLs, however, are not as large because the conv layers dominate the computational complexity. Other types of NNs, such as restricted Boltzmann machines, have higher prominence of CLs than CNNs and would thus benefit more from our approach. Table 2 shows the overall memory and computational gains obtained from pre-defining CLs to be sparse for our networks. The number of SCL parameters (params) are calculated by taking the minimum overall CL density at which there is no accuracy loss. Note that the number of operations (ops) for CLs is nearly the same as their number of parameters, hence are not explicitly shown.

Table 2: Savings in some of our NN architectures due to pre-defined sparsity

| Net | CLs / Total Layers | Conv Params (M) | Conv Ops (B) | FC CL Params (M) | Sparse CL Params (M) | Overall Param % Reduction | Overall Op % Reduction |
|---|---|---|---|---|---|---|---|
| Morse CL | 2/2 | 0 | 0 | 0.131 | 0.066 | 50 | 50 |
| MNIST CL ($x = 224$) | 2/2 | 0 | 0 | 0.178 | 0.089 | 50 | 50 |
| MNIST conv+2CLs | 2/6 | 0.05 | 0.1 | 2.47 | 0.06 | 95.63 | 18.29 |
| CIFAR10 conv+2CLs | 2/17 | 1.15 | 0.15 | 2.11 | 0.005 | 64.63 | 1.35 |
| CIFAR100 conv+2CLs | 2/17 | 1.15 | 0.15 | 2.16 | 0.02 | 64.76 | 1.38 |
| CIFAR10 conv+3CLs | 3/18 | 1.15 | 0.15 | 2.23 | 0.009 | 65.83 | 1.43 |
| CIFAR100 conv+3CLs | 3/18 | 1.15 | 0.15 | 2.26 | 0.013 | 65.99 | 1.45 |

## 2.4 DISTRIBUTING INDIVIDUAL JUNCTION DENSITIES

Note that the Morse code network has symmetric junctions since each will have $64 \times 1024 = 65,536$ weights to give a total of 131,072 FCL weights. Consider an example where overall density of 50% (i.e. 65,536 total SCL weights) is desired. This can be achieved in multiple ways, such as making both junctions 50% dense, i.e. 32,768 weights in each. Here we explore if individual junction densities contribute equally to network performance.

Figures 3(b) and (c) sweep junction 1 and 2 connectivity densities on the x-axis such that the resulting overall density is fixed at 25% for (b) and 50% for (c). The black vertical line denotes where the densities are equal. Note that peak performance in both cases is achieved to the left of the black line, such as in (c) where junction 2 is 75% dense and junction 1 is 25% dense. This suggests that later junctions need more connections than earlier ones. See Appendix Section 5.1 for more details.

## 3 CONNECTIVITY PATTERNS

We now introduce adjacency matrices to describe junction connection patterns. Let $A_i \in \{0, 1\}^{N_{i+1} \times N_i}$ be the (simplified) adjacency matrix of junction $i$, such that element $[A_i]_{j,k}$ indicates whether there is a connection between the $j$th right neuron and $k$th left neuron. $A_i$ will have $fi_i$ 1's on each row and $fo_i$ 1's on each column. These adjacency matrices can be multiplied to yield the ef-

fective connection pattern between any 2 junctions $X$ and $Y$, i.e. $A_{X:Y} = \prod_{i=Y}^{X} A_i \in \mathbb{Z}_{\geq 0}^{N_{Y+1} \times N_X}$, where element $[A_{X:Y}]_{j,k}$ denotes the number of paths from the $k$th neuron in layer $X$ to the $j$th neuron in layer $(Y+1)$. For the special case where $X = 1$ and $Y = J$ (total number of junctions), we obtain the input-output adjacency matrix $A_{1:J}$. As a simple example, consider the $(8, 4, 4)$ network shown in Fig. 4 where $fo_1 = 1$ and $fo_2 = 2$, which implies that $fi_1 = fi_2 = 2$. $A_1$ and $A_2$ are adjacency matrices of single junctions. We obtain the input-output adjacency matrix $A_{1:2} = A_2 A_1$, equivalent $fo_{1:2} = fo_1 fo_2 = 2$, and equivalent $fi_{1:2} = fi_1 fi_2 = 4$. Note that this equivalent junction 1:2 is only an abstract concept that aids visualizing how neurons connect from the inputs to the outputs. It has no relation to the overall network density.

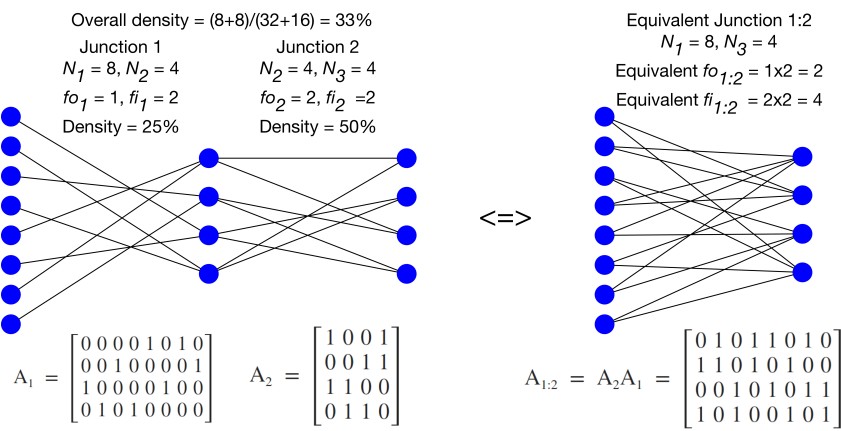

Figure 4: An example of adjacency matrices and equivalent junctions.

We now attempt to characterize the quality of a sparse connection pattern, i.e. we try to find the best possible way to connect neurons to optimize performance. Since sparsity gives good performance, we hypothesize that there exists redundancy / correlated information between neurons. Intuitively, we assume that left neurons of a junction can be grouped into *windows* depending on the dimensionality of the left layer output. For example, the input layer in an MNIST CL only network would have 2D windows, each of which might correspond to a fraction of the image, as shown in Fig. 5(a). When outputs from a CL have an additional dimension for features, such as in CIFAR or the MNIST conv network, each window is a cuboid capturing fractions of both spatial extent and features, as shown in Fig. 5(b). Given such windows, we will try to maximize the number of left windows to which each right neuron connects, the idea being that each right neuron should get some information from all portions of the left layer in order to capture global view. To realize the importance of this, consider the MNIST output neuron representing digit 2. Let's say the sparse connection pattern is such that when the connections to output 3 are traced back to the input layer, they all come from the top half of the image. This would be undesirable since the top half of an image of a 2 can be mistaken for a 3. A good sparse connection pattern will try to avoid such scenarios by spreading the connections to any right neuron across as many input windows as possible. The problem can also be mirrored so that every left neuron connects to as many different right windows as possible. This ensures that local information from left neurons is spread to different parts of the right layer. The grouping of right windows will depend on the dimensionality of the input to the right layer.

The window size is chosen to be the minimum possible such that the ideal number of connections from or to it remains integral. The example from Fig. 4 is reproduced in Fig. 6. Since $fi_1 = 2$, the inputs must be grouped into 2 windows so that ideally 1 connection from each reaches every hidden neuron. If instead the inputs are grouped into 4 windows, the ideal number would be half of a connection, which is not achievable. In order to achieve the minimum window size, we let the number of left windows be $fi$ and the number of right windows be $fo$. So in junction $i$, the number of neurons in each left and right window is $N_i/fi_i$ and $N_{i+1}/fo_i$, respectively. Then we construct left- and right-window adjacency matrices $A_i^{w_{il}} \in \mathbb{Z}_{\geq 0}^{N_{i+1} \times fi_i}$ and $A_i^{w_{ir}} \in \mathbb{Z}_{\geq 0}^{fo_i \times N_i}$ by summing up entries of $A_i$ as shown in Fig. 5(c). The window adjacency matrices describe connectivity between windows and neurons on the opposite side. Ideally, every window adjacency matrix for a single junction should be the all 1s matrix, which signifies exactly 1 connection from every window to

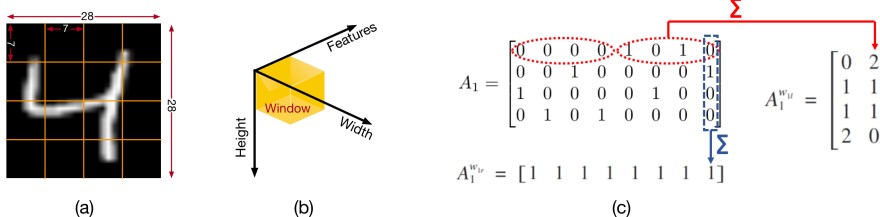

Figure 5: (a) Example of 16 2D windows for an MNIST input image. (b) Example of 3D windows when the output from a layer also includes features. (c) Construction of window adjacency matrices.

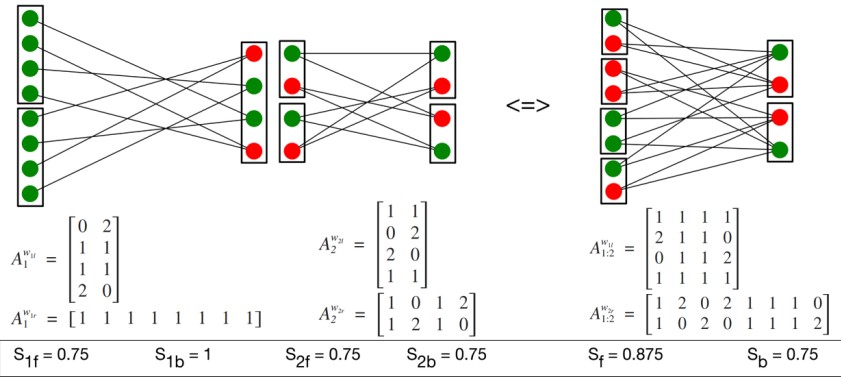

Figure 6: Window adjacency matrices and scatter. Green neurons indicate ideal connectivity. The hidden layer is split into 2 to show separate constructions of $A_1^{w_{1r}}$ and $A_2^{w_{2l}}$
.

every neuron on the opposite side. Note that these matrices can also be constructed for multiple junctions, i.e. $A_{X:Y}^{w_{Xl}}$ and $A_{X:Y}^{w_{Yr}}$, by multiplying matrices for individual junctions. See Appendix Section 5.2 for more discussion.

## 3.1 SCATTER

Scatter is a proxy for the performance of a NN. It is useful because it can be computed in a fraction of a second and used to predict how good or bad a sparse network is without spending time training it. To compute scatter, we count the number of entries greater than or equal to 1 in the window adjacency matrix. If a particular window gets more than its fair share of connections to a neuron on the opposite side, then it is depriving some other window from getting its fair share. This should not be encouraged, so we treat entries greater than 1 the same as 1. Scatter is the average of the count, i.e. for junction $i$:

$$S_{if} = \frac{1}{fi_i N_{i+1}} \sum_{j=1}^{N_{i+1}} \sum_{k=1}^{fi_i} \mathbb{I}\Big([A_i^{w_{il}}]_{j,k} \geq 1\Big), \qquad S_{ib} = \frac{1}{N_i fo_i} \sum_{j=1}^{fo_i} \sum_{k=1}^{N_i} \mathbb{I}\Big([A_i^{w_{ir}}]_{j,k} \geq 1\Big). \quad (1)$$

Subscripts $f$ and $b$ denote forward (left windows to right neurons) and backward (right neurons to left windows), indicating the direction of data flow. As an example, we consider $A_1^{w_{1l}}$ in Fig. 6, which has a scatter value $S_{1f} = 6/8 = 0.75$. The other scatter values can be computed similarly to form the scatter vector $\bar{S} = [S_{1f}, S_{1b}, S_{2f}, S_{2b}, S_f, S_b]$, where the final 2 values correspond to junction 1:2. Notice that $\bar{S}$ will be all 1s for FCLs, which is the ideal case. Incorporating sparsity leads to reduced $\bar{S}$ values. The final scatter metric $S \in [0, 1]$ is the minimum value in $\bar{S}$, i.e. 0.75 for Fig. 6. Our experiments indicate that any low value in $\bar{S}$ leads to bad performance, so we picked the critical minimum value.

## 3.2 ANALYSIS AND RESULTS OF SCATTER

We ran experiments to evaluate scatter using a) the Morse CL only network with $fo = 128, 8$, b) an MNIST CL only network with $(1024, 64, 16)$ neuron configuration and $fo = 1, 4$, and c) the 'conv+2CLs' CIFAR10 network with $fo = 1, 2$. We found that high scatter indicates good performance and the correlation is stronger for networks where CLs have more importance, i.e. CL only networks as opposed to conv. This is shown in the performance vs. scatter plots in Fig. 7, where (a) and (b) show the performance predicting ability of scatter better than (c). Note that the random connection patterns used so far have the highest scatter and occur as the rightmost points in each subfigure. The other points are obtained by specifically planning connections. We found that when 1 junction was planned to give corresponding high values in $\bar{S}$, it invariably led to low values for another junction, leading to a low $S$. This explains why random patterns generally perform well.

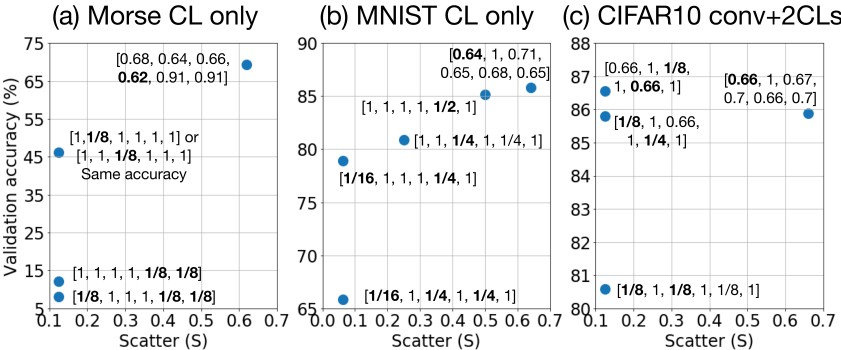

Figure 7: Network performance vs. scatter for CL only networks of (a) Morse (b) MNIST, and convolutional network with 2 CL junctions of (c) CIFAR10. All minimum values that need to be considered to differentiate between connection patterns are bolded.

$\bar{S}$ is shown alongside each point. When $S$ is equal for different connection patterns, the next minimum value in $\bar{S}$ needs to be considered to differentiate the networks, and so on. Considering the Morse results, the leftmost 3 points all have $S = \frac{1}{8}$, but the number of occurrences of $\frac{1}{8}$ in $\bar{S}$ is 3 for the lowest point (8% accuracy), 2 for the second lowest (12% accuracy) and 1 for the highest point (46% accuracy). For the MNIST results, both the leftmost points have a single minimum value of $\frac{1}{16}$ in $\bar{S}$, but the lower has two occurrences of $\frac{1}{4}$ while the upper has one.

We draw several insights from these results. Firstly, although we defined $S$ as a single value for convenience, there may arise cases when other (non-minimum) elements in $\bar{S}$ are important. Secondly, perhaps contrary to intuition, the concept of windows and scatter is important for all CLs, not simply the first. As shown in Fig. 7a), a network with $S_{1b} = \frac{1}{8}$ performs equally poorly as a network with $S_{2f} = \frac{1}{8}$. Thirdly, scatter is a sufficient metric for performance, not necessary. A network with a high $S$ value will perform well, but a network with a slightly lower $S$ than another cannot be conclusively dismissed as being worse. But if a network has multiple low values in $\bar{S}$, it should be rejected. Finally, carefully choosing which neurons to group in a window will increase the predictive power of scatter. *A priori* knowledge of the dataset will lead to better window choices.

## 4 CONCLUSION AND FUTURE WORK

This paper discusses the merits of pre-defining sparsity in CLs of neural networks, which leads to significant reduction in parameters without performance loss. In general, the smaller the fraction of CLs in a network, the more redundancy there exists in their parameters. If we can achieve similar results (i.e., 0.2% density) on Alexnet for example, we would obtain 95% reduction in overall parameters. Coupled with hardware acceleration designed for pre-defined sparse networks, we believe our approach will lead to more aggressive exploration of network structure. Network connectivity can be guided by the scatter metric, which is closely related to performance, and by optimally distributing connections across junctions. Future work would involve extension to conv layers, since recent CNNs have lower values for the ratio of number of CLs to number of conv layers.

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

# 5 APPENDIX

## 5.1 MORE ON DISTRIBUTING INDIVIDUAL JUNCTION DENSITIES

Section 2.4 showed that when overall CL density is fixed, it is desirable to make junction 2 denser than junction 1. It is also interesting to note, however, that performance falls off more sharply when junction 1 density is reduced to the bare minimum as compared to treating junction 2 similarly. This is not shown in Fig. 3 due to space constraints. We found that when junction 1 had the minimum possible density and junction 2 had the maximum possible while still satisfying the fixed overall, the accuracy was about 36% for both subfigures (b) and (c). When the densities were flipped, the accuracies were 67% for subfigure (b) and 75% for (c) in Figure 3.

## 5.2 DENSE CASES OF WINDOW ADJACENCY MATRICES

As stated in Section 3.1, window output matrices for several junctions can be constructed by multiplying the individual matrices for each component junction. Consider the Morse network as described in Section 3.2. Note that $fo_{1:2} = 128 \times 8 = 1024$ and $fi_{1:2} = 8 \times 128 = 1024$. Thus, for the equivalent junction 1:2 which has $N_1 = 64$ left neurons and $N_3 = 64$ right neurons, we have $fo_{1:2} > N_3$ and $fi_{1:2} > N_1$. So in this case the number of neurons in each window will be rounded up to 1, and both the ideal window adjacency matrices $A_{1:2}^{w_{1l}}$ and $A_{1:2}^{w_{2r}}$ will be all 16's matrices since the ideal number of connections from each window to a neuron on the opposite side is $1024/64 = 16$. This is a result of the network having sufficient density so that several paths exist from every input neuron to every output neuron.

## 5.3 POSSIBLE REASONS FOR SCLs CONVERGING FASTER THAN FCLs

Training a neural network is essentially an exercise in finding the minimum of the cost function, which is a function of all the network parameters. The graph for cost as a function of parameters may have saddle points which masquerade as minima. It could also be poorly conditioned, wherein the gradient of cost with respect to two different parameters have widely different magnitudes, making simultaneous optimization difficult. These effects are non-idealities and training the network often takes more time because of the length of the trajectory needed to overcome these and arrive at the optimum point. The probability of encountering these non-idealities increases as the number of network parameters increase, i.e. less parameters leads to a higher ratio of minima : saddle points, which can make the network converge faster. We hypothesize that SCLs train faster than FCLs due to the former having fewer parameters.

