# OpenReview forum: "Characterizing Sparse Connectivity Patterns in Neural Networks"
_ICLR.cc/2018/Conference — Reject_

### Official Review · AnonReviewer2 · 2017-11-27
**hard to follow, confused about many points**

**Rating:** 4
**Confidence:** 3

**Review:**

This paper examines sparse connection patterns in upper layers of convolutional image classification networks.  Networks with very few connections in the upper layers are experimentally determined to perform almost as well as those with full connection masks.  Heuristics for distributing connections among windows/groups and a measure called "scatter" are introduced to construct the connectivity masks, and evaluated experimentally on CIFAR-10 and -100, MNIST and Morse code symbols.

While it seems clear in general that many of the connections are not needed and can be made sparse (Figures 1 and 2), I found many parts of this paper fairly confusing, both in how it achieves its objectives, as well as much of the notation and method descriptions.  I've described many of the points I was confused by in more detailed comments below.


Detailed comments and questions:


The distribution of connections in "windows" are first described to correspond to a sort of semi-random spatial downsampling, to get different views distributed over the full image.  But in the upper layers, the spatial extent can be very small compared to the image size, sometimes even 1x1 depending on the network downsampling structure.  So are do the "windows" correspond to spatial windows, and if so, how?  Or are they different (maybe arbitrary) groupings over the feature maps?

Also a bit confusing is the notation "conv2", "conv3", etc.  These names usually indicate the name of a single layer within the network (conv2 for the second convolutional layer or series of layers in the second spatial size after downsampling, for example).  But here it seems just to indicate the number of "CL" layers: 2.  And p.1 says that the "CL" layers are those often referred to as "FC" layers, not "conv" (though they may be convolutionally applied with spatial 1x1 kernels).

The heuristic for spacing connections in windows across the spatial extent of an image makes intuitive sense, but I'm not convinced this will work well in all situations, and may even be sub-optimal for the examined datasets.  For example, to distinguish MNIST 1 vs 7 vs 9, it is most important to see the top-left:  whether it is empty, has a horizontal line, or a loop.  So some regions are more important than others, and the top half may be more important than an equally spaced global view.  So the description of how to space connections between windows makes some intuitive sense, but I'm unclear on whether other more general connections might be even better, including some that might not be as easily analyzed with the "scatter" metric described.

Another broader question I have is in the distinction between lower and upper layers (those referred to as "feature extracting" and "classification" in this paper).  It's not clear to me that there is a crisply defined difference here (though some layers may tend to do more of one or the other function, such as we might interpret).  So it seems that expanding the investigation to include all layers, or at least more layers, would be good:  It might be that more of the "classification" function is pushed down to lower layers, as the upper layers are reduced in size.  How would they respond to similar reductions?

I'm also unsure why on p.6 MNIST uses 2d windows, while CIFAR uses 3d --- The paper mentions the extra dimension is for features, but MNIST would have a features dimension as well at this stage, I think?  I'm also unsure whether the windows are over spatial extent only, or over features.

---

> ### Author Response · Authors · 2018-01-03
> **Clarifications made and concerns addressed - Part 1**
>
> Thank you for reviewing the paper and making insightful comments. We have addressed them and revised the paper to the best of our ability, and hope that your concerns are sufficiently addressed.
>
>
> Question: The distribution of connections in "windows" are first described to correspond to a sort of semi-random spatial downsampling, to get different views distributed over the full image.  But in the upper layers, the spatial extent can be very small compared to the image size, sometimes even 1x1 depending on the network downsampling structure.  So are do the "windows" correspond to spatial windows, and if so, how?  Or are they different (maybe arbitrary) groupings over the feature maps?
>
> Answer: As we have clarified in Section 3, windows in the left (right) layer of a junction will correspond to the dimensionality of the output (input) of that layer. For example, the input layer in an MNIST CL only network would have 2D windows, each of which might correspond to a fraction of the image, as shown in Fig. 5(a). When inputs to a CL have an additional dimension for features, such as in CIFAR or the MNIST conv network, each window is a cuboid capturing fractions of both spatial extent and features, as shown in Fig. 5(b).
> For the spatial windows, nearby pixels have correlated information, so we hypothesize that each right neuron needs only 1 connection from each such spatial window. For different feature maps, the extent of correlation is unknown. So in their case, the grouping is arbitrary.
>
>
> Question: Also a bit confusing is the notation "conv2", "conv3", etc.  These names usually indicate the name of a single layer within the network (conv2 for the second convolutional layer or series of layers in the second spatial size after downsampling, for example).  But here it seems just to indicate the number of "CL" layers: 2.  And p.1 says that the "CL" layers are those often referred to as "FC" layers, not "conv" (though they may be convolutionally applied with spatial 1x1 kernels).
>
> Answer: We have made 2 changes to clear up the notation:
> a) Layers which are conventionally fully connected, i.e. those which we aim to make sparse, are now being called connected layers (CLs). Fully connected layers (FCLs) and sparsely connected layers (SCLs) that we have proposed are both special cases of CLs.
> b) The notation ‘conv2’ has been changed to ‘conv+2CLs’, and similarly for ‘conv3’
>
>
> Question: The heuristic for spacing connections in windows across the spatial extent of an image makes intuitive sense, but I'm not convinced this will work well in all situations, and may even be sub-optimal for the examined datasets.  For example, to distinguish MNIST 1 vs 7 vs 9, it is most important to see the top-left:  whether it is empty, has a horizontal line, or a loop.  So some regions are more important than others, and the top half may be more important than an equally spaced global view.  So the description of how to space connections between windows makes some intuitive sense, but I'm unclear on whether other more general connections might be even better, including some that might not be as easily analyzed with the "scatter" metric described.
>
> Answer: The main value of scatter lies in it being an indicator, i.e. if a network has high scatter, it will definitely perform well, and if there are multiple low values in the scatter bar vector, performance will generally be poor. But the metric has its limitations, such as uncertainty regarding exact bounds which guarantee a certain level of network performance. The predictive power of scatter is largely influenced by the chosen windows. We are currently working on improvements, such as using a priori dataset knowledge on how to choose windows and decide correlation between different spatial sections of an image and its features.

---

> > ### Author Response · Authors · 2018-01-03
> > **Clarifications made and concerns addressed - Part 2**
> >
> > Continued from the previous comment...
> >
> >
> > Question: Another broader question I have is in the distinction between lower and upper layers (those referred to as "feature extracting" and "classification" in this paper).  It's not clear to me that there is a crisply defined difference here (though some layers may tend to do more of one or the other function, such as we might interpret).  So it seems that expanding the investigation to include all layers, or at least more layers, would be good:  It might be that more of the "classification" function is pushed down to lower layers, as the upper layers are reduced in size.  How would they respond to similar reductions?
> >
> > Answer: This is a very good point - the exact function of different layers is not so clearly demarcated in very deep networks. As mentioned in the paper conclusion, the next step is to extend sparsity methodologies to convolutional layers. But note that conv layers are already sparse by definition (since a neuron in a layer connects to only a few in another layer). Hence we believe that the scope for significantly reducing parameters without adversely affecting performance is far greater in fully connected layers.
> >
> >
> > Question: I'm also unsure why on p.6 MNIST uses 2d windows, while CIFAR uses 3d --- The paper mentions the extra dimension is for features, but MNIST would have a features dimension as well at this stage, I think?  I'm also unsure whether the windows are over spatial extent only, or over features.
> >
> > Answer: As clarified in the answer to your first question and in Section 3 and Figure 5 in the paper, the dimension of windows in a left (right) layer of a junction is the same as the dimension of its output (input). So for example, the input layer for an MNIST CL only network will have 2D windows, while the 1st CL in an MNIST convolutional network will have 3D windows.

---

### Official Review · AnonReviewer1 · 2017-12-03
**Sparse networks have fewer parameters without potentially sacrificing performance**

**Rating:** 5
**Confidence:** 3

**Review:**

The authors propose reducing the number of parameters learned by a deep network by setting up sparse connection weights in classification layers. Numerical experiments show that such sparse networks can have similar performance to fully connected ones. They introduce a concept of “scatter” that correlates with network performance. Although  I found the results useful and potentially promising, I did not find much insight in this paper.
It was not clear to me why scatter (the way it is defined in the paper) would be a useful performance proxy anywhere but the first classification layer. Once the signals from different windows are intermixed, how do you even define the windows?
Minor
Second line of Section 2.1: “lesser” -> less or fewer

---

> ### Author Response · Authors · 2018-01-03
> **Clarifications made and concerns addressed**
>
> Thank you for reviewing the paper and making insightful comments. We have addressed them and revised the paper to the best of our ability, and hope that your concerns are sufficiently addressed.
>
>
> Comment: Although I found the results useful and potentially promising, I did not find much insight in this paper. It was not clear to me why scatter (the way it is defined in the paper) would be a useful performance proxy anywhere but the first classification layer.
>
> Response:
> a) Let me explain by giving an example of a network with 3 CLs, connected as shown in this figure: https://drive.google.com/file/d/1tTGtdeyAwPvzbQ2YWeTQicDzm1RPn38q/view?usp=sharing
> If we compute all the scatter vector values, S_f and S_b will be good because every output neuron is connected to every input neuron, i.e. the input-to-output connectivity is good. But this is not a good network because 2 of the 3 hidden neurons are being wasted and can be removed. The problem with this network is captured by the other scatter values S_1f, S_1b, S_2f and S_2b, which will be poor. This is why all the values in the scatter vector need to be considered, since some low values may lead to performance degradation, as shown in Fig. 7.
> This is a toy example used for demonstration, but we simulated a larger example using a similar approach and obtained inferior performance. We hope this serves to explain why intermediate hidden layer connectivity is important.
>
> b) It has been shown in the literature that non-linearity is required in neural networks to improve their approximation capabilities, particularly for problems which are not linearly separable. Such non-linearity is captured by ReLU activations in the hidden layers. If we just take the scatter values involving the 1st CL, or just the scatter values of the input-output equivalent junction, we ignore the importance of non-linearity effect introduced by the hidden layers. As shown in Fig. 7a), a network where the only low scatter value is S_1b = ⅛ performs equally badly as a network where the only low scatter value is S_2f = ⅛, even though the latter has good connectivity in the 1st CL.
>
>
> Question: Once the signals from different windows are intermixed, how do you even define the windows?
>
> Answer: As shown in Fig. 6 (previously fig. 5), the windows in the hidden layers are groups of adjacent neurons. We follow this approach based on the assumption that we need good mixing overall, i.e. both individual junctions 1 and 2, need to be mixed, as well as the equivalent junction 1:2. This assumption is justified by the reasoning from the response to the previous comment. Thus, the entire scatter vector is important. This insight on scatter, along with a few others, have been included in Section 3.2 of the revised paper.
>
>
> Comment: Minor Second line of Section 2.1: “lesser” -> less or fewer
>
> Response: Thank you for pointing this out. We changed the word to ‘fewer’.

---

### Official Review · AnonReviewer3 · 2017-12-18
**Deep compression alternative?**

**Rating:** 4
**Confidence:** 3

**Review:**

The paper seems to claims that
1) certain ConvNet architectures, particularly AlexNet and VGG, have too many parameters,
2) the sensible solution is leave the trunk of the ConvNet unchanged, and to randomly sparsify the top-most weight matrices.
I have two problems with these claims:
1) Modern ConvNet architectures (Inception, ResNeXt, SqueezeNet, BottleNeck-DenseNets and ShuffleNets) don't have large fully connected layers.
2) The authors reject the technique of 'Deep compression' as being impractical. I suspect it is actually much easier to use in practice as you don't have to a-priori know the correct level of sparsity for every level of the network.

p3. What does 'normalized' mean? Batch-norm?
p3. Are you using an L2 weight penalty? If not, your fully-connected baseline may be unnecessarily overfitting the training data.
p3. Table 1. Where do the choice of CL Junction densities come from? Did you do a grid search to find the optimal level of sparsity at each level?
p7-8. I had trouble following the left/right & front/back notation.
p8. Figure 7. How did you decide which data points to include in the plots?

---

> ### Author Response · Authors · 2018-01-03
> **Clarifications made and concerns addressed**
>
> Thank you for reviewing the paper and making insightful comments. We have addressed them and revised the paper to the best of our ability, and hope that your concerns are sufficiently addressed.
>
>
> Comment: Modern ConvNet architectures (Inception, ResNeXt, SqueezeNet, BottleNeck-DenseNets and ShuffleNets) don't have large fully connected layers.
>
> Response: We agree that some recent CNN architectures have attempted to reduce the number of FC layers. However, ResNeXt, DenseNet and ShuffleNet all have 1 final softmax FC layer, which account for approximately 3%, 28-48% and 12-30% of the overall parameters as per our calculations (the ranges indicate different DenseNet and ShuffleNet architectures). For Inception, FC parameters account for 74% in auxiliary classifier 0, 34% in auxiliary classifier 1, and 11% in the main classifier. Although these numbers are less than other architectures, we believe there are still significant savings to be achieved by reducing the density of these FC layers as per our other experiments given in the paper.
> Note that we assume the typical scenario where the outputs of the final convolutional layer are flattened before getting fully connected to the softmax classifier.
>
> SqueezeNet does not mention the use of any FC layer. Our ongoing work, as mentioned in the conclusion section of our submission, is exploring techniques to sparsify CNNs. Some methods already exist, such as depthwise convolutions (XCeption, Shufflenet) and grouping convolutions (AlexNet, ResNeXt, ShuffleNet). Note that using these methods to reduce conv params means that the fraction of FC params goes up, which further justifies our methods to sparsify FC layers.
>
>
> Comment: The authors reject the technique of 'Deep compression' as being impractical. I suspect it is actually much easier to use in practice as you don't have to a-priori know the correct level of sparsity for every level of the network.
>
> Response: One of our goals, as mentioned in the paper, is hardware acceleration of neural networks. In particular, some of the works we have cited such as Dey et al. (2017a;b) have leveraged pre-defined sparsity to simplify the memory and computational footprint of neural network hardware architectures capable of on-chip training and inference. Deep Compression uses post-training sparse methods such as pruning and quantization, which are unsuited for on-chip training. This is because the entire (non-sparse) architecture needs to be used for training, and then additional computation done to reduce parameters. This is why we propose starting off with a sparse architecture.
>
>
> Question: p3. What does 'normalized' mean? Batch-norm?
>
> Response: Yes, we are referring to batch-normalization. We have modified the paper to clarify this.
>
>
> Question: p3. Are you using an L2 weight penalty? If not, your fully-connected baseline may be unnecessarily overfitting the training data.
>
> Response: Yes we experimented with different values for L2 weight penalty coefficient and picked the optimum values. The paper has been revised to indicate this.
>
> Question: p3. Table 1. Where do the choice of CL Junction densities come from? Did you do a grid search to find the optimal level of sparsity at each level?
>
> Response: Yes, we did a grid search. To simplify the search, we focused more on architectures with higher junction densities in the later (closer to output) layers. This is in accordance with our findings in Section 2.4. The paper has been revised to indicate this.
>
> Comment: p7-8. I had trouble following the left/right & front/back notation.
>
> Response: Layers closer to the input are ‘left’ and those closer to the output are ‘right’. Left to right indicates forward. Right to left indicates backward. We have modified the paper to explicitly mention this. For example, S_1f refers to the scatter value when going forward in junction 1, i.e. windows are formed in the input layer to the left, and data flows from them to neurons in the hidden layer to the right.
>
>
> Question: p8. Figure 7. How did you decide which data points to include in the plots?
>
> Response: As mentioned in Section 3.2, we tried random and planned connection patterns. Several random connection patterns led to similar values for scatter, so we included only 1 of them. For the planned points, we distributed the connections in such a way so that certain junctions had perfect window-to-neuron connectivity, i.e. some values in the scatter vector would be 1. As mentioned, this invariably led to some other values being very low. The points included in the plots serve to highlight how all the scatter vector values are important, i.e. how a single low value can lead to bad performance.

---

### Author Response · Authors · 2018-01-04
**Revision uploaded - major changes made are listed here**

Terminology changed - CL refers to connected layers, of which fully connected layer (FCL) is a special case. Our proposed technique of pre-defining a layer to be sparse by not having most of the connections will lead to a sparsely connected layer (SCL), which is also a special case of CL.

Several portions of Section 2 have been changed to indicate that we used L2 regularization, batch normalization and grid search to decide the optimum junction densities for different networks.

The MNIST CL only simulations are done more extensively using several networks having varying number of hidden layers. Fig 2c has been bolstered as a result and new insights offered in Section 2.2.2, which indicate that large sparse networks perform better than small dense networks.

Section 3 has been clarified to indicate the nature of windows in different junctions. In particular, left windows in a junction correspond to the dimensionality of the output of the left layer, while right windows in a junction correspond to the dimensionality of the input of the right layer. New subfigures (5a and 5b) have been added to illustrate this.

The dimensionality of the left- and right- window adjacency matrices has been corrected and Equation 1 fixed.

Insights on scatter are offered in Section 3.2

A new appendix section 5.3 has been added to explain possible reasons behind SCLs converging faster than FCLs, as shown in Fig 1.

Other minor changes have been made which are not listed here due to space constraints. Some of these are in response to reviewers' comments, such as changing the terminology 'conv2' to 'conv+2CLs'. More details can be found in the comments below. Other changes are done to tighten the language and make the paper fit in 8 pages.

---

### Decision · Program_Chairs · 2018-01-29
**ICLR 2018 Conference Acceptance Decision**

**Decision:**

Reject

**Comment:**

The paper received weak scores: 4,4,5. R2 complained about clarity. R3's point about the lack of fully connected layers in current SOA deepnets is very valid and the authors response far from convincing. Unfortunately the major revision provided by the authors was not commented on by the reviewers, but many of the major shortcomings of the work still remain.
Generally, the paper is below the acceptance threshold, so cannot be accepted.